# Integration of Genome and Epigenetic Testing in the Diagnostic Evaluation of Developmental Delay: Differentiating Börjeson–Forssman–Lehmann (BFLS) and White–Kernohan (WHIKERS) Syndromes

**DOI:** 10.3390/genes16080933

**Published:** 2025-08-04

**Authors:** Keri Ramsey, Supraja Prakash, Jennifer Kerkhof, Bekim Sadikovic, Susan White, Marcus Naymik, Jennifer Sloan, Anna Bonfitto, Newell Belnap, Meredith Sanchez-Castillo, Wayne Jepsen, Matthew Huentelman, Saunder Bernes, Vinodh Narayanan, Shagun Kaur

**Affiliations:** 1Center for Rare Childhood Disorders, Translational Genomics Research Institute (TGen), Phoenix, AZ 85004, USA; ramsey.keri@mayo.edu (K.R.); mnaymik@tgen.org (M.N.); jsloan@tgen.org (J.S.); abonfitto@tgen.org (A.B.); nbelnap@tgen.org (N.B.); msanchez-castillo@tgen.org (M.S.-C.); wjepsen@tgen.org (W.J.); mhuentelman@tgen.org (M.H.); 2Phoenix Children’s Hospital, Phoenix, AZ 85016, USA; sprakash@phoenixchildrens.com (S.P.); sbernes@phoenixchildrens.com (S.B.); skaur2@phoenixchildrens.com (S.K.); 3Verspeeten Clinical Genome Center, London Health Sciences Center, London, ON N6A 5W9, Canada; jennifer.kerkhof@lhsc.on.ca (J.K.); bekim.sadikovic@lhsc.on.ca (B.S.); 4Department of Pathology and Laboratory Medicine, Western University, London, ON N6A 5C1, Canada; 5Victorian Clinical Genetics Services, Murdoch Children’s Research Institute, Melbourne, VIC 3052, Australia; sue.white@vcgs.org.au; 6Department of Paediatrics, Univerwsity of Melbourne, Melbourne, VIC 3010, Australia; 7Department of Child Health, University of Arizona College of Medicine-Phoenix, Phoenix, AZ 85004, USA

**Keywords:** DNA methylation, Epigenetics, VUS clarification, genome sequencing, intellectual disability, BFLS, WHIKERS

## Abstract

**Background**: More than 1500 genes are associated with developmental delay and intellectual disability, with variants in many of these genes contributing to a shared phenotype. The discovery of variants of uncertain significance (VUS) found in these genes during genetic testing can lead to ambiguity and further delay in diagnosis and medical management. Phenotyping, additional genetic testing, and functional studies can all add valuable information to help reclassify these variants. Here we demonstrate the clinical utility of epigenetic signatures in prioritizing variants of uncertain significance in genes associated with developmental delay (DD) and intellectual disability (ID). **Methods**: Genome sequencing was performed in a male with developmental delay. He was found to have VUSs in both *PHF6* and *DDB1* genes, linked with Börjeson–Forssman–Lehmann syndrome (BFLS) and White–Kernohan syndrome (WHIKERS), respectively. These two disorders share a similar phenotype but have distinct inheritance patterns and molecular pathogenic mechanisms. DNA methylation profiling (DNAm) of whole blood was performed using the clinically validated EpiSign assay. **Results**: The proband’s methylation profile demonstrated a strong correlation with the BFLS methylation signature, supporting the *PHF6* variant as a likely cause of his neurodevelopmental disorder. **Conclusions**: Epigenetic testing for disorders with distinct methylation patterns can provide diagnostic utility when a patient presents with variants of uncertain significance in genes associated with developmental delay. Epigenetic signatures can also guide genetic counselling and family planning.

## 1. Introduction

White–Kernohan syndrome (WHIKERS) and Börjeson–Forssman–Lehmann syndrome (BFLS) are rare neurodevelopmental disorders, each with unique molecular mechanisms yet sharing notable clinical similarities [1,2]. WHIKERS is caused by heterozygous variants in the *DDB1* gene, while BFLS is an X-linked disorder caused by hemizygous variants (males) and heterozygous variants (some females) in the *PHF6* gene. WHIKERS and BFLS have an overlapping clinical phenotype characterized by developmental delay/intellectual disability, hypotonia, obesity, and dysmorphic physical features. Despite these distinct genetic causes, recent epigenetic studies have highlighted overlapping DNA methylation patterns between these two syndromes, providing new insights into their shared molecular signatures.

The damage-specific DNA-binding protein 1 (DDB1, encoded by the *DDB1 gene*) functions as part of the CUL4-DDB1 ubiquitin E3 ligase complex which plays a role in the nucleotide excision repair pathway [1]. Only eight individuals with WHIKERS have been described in the literature and display hypotonia, mild-moderate developmental delay and intellectual disability, distinctive facial features, and anomalies of the hands and feet [1]. Brachydactyly is common and some individuals have had cutaneous syndactyly of the toes. Dysmorphic features include large and fleshy ears and earlobes, horizonal or slightly bowed eyebrows, mild narrowing of the palpebral fissures, full cheeks, and a short nose. Variants in *DDB1* reported in patients with WHIKERS are predominantly missense, with a single patient having an in-frame deletion located in the first mono-functional DNA-alkylating methyl methanesulfonate (MMS1) domain of the protein. ClinVar lists one variant p.(E785K) found between the MMS1 and the cleavage and polyadenylation specificity factor (CPSF) domain. This variant (ID 3384139) is present in one heterozygous individual in gnomAD (v.4.1.0) [3].

Börjeson–Forssman–Lehmann syndrome (BFLS) was first described by Dr. Börjeson and colleagues in 1962 in a family with three males with intellectual disability, epilepsy, hypogonadism, obesity, swelling of the subcutaneous tissue of the face, narrow palpebral fissures, and large ears [4]. Three females in the family were also characterized with ‘moderate mental deficiency’. In 2002, 40 years later, Lower and colleagues identified variants in the plant homeodomain finger protein 6 gene (*PHF6)* located on the X chromosome in nine families with a BFLS diagnosis [2]. There are fewer than 100 individuals reported in the literature. In addition, somatic variants in the *PHF6* gene have been associated with blood cancers (mostly T-cell acute lymphoblastic leukemia), and hematological malignancy has been described in several individuals with BFLS [5]. Although BFLS mostly affects males, heterozygous females, while rare, may have moderate to severe intellectual disability, facial dysmorphism, and differences in the teeth, fingers, and toes [6]. This phenotype variability may be attributed to the severity of the damaging effect of the variant itself, X-inactivation pattern, or mosaicism [6]. PHF6 protein levels in patient cell lines were greatly reduced in individuals with missense variants (p.C45Y, p.C99F) but were normal or even elevated in those with deletions (p.D333del, p.E337del) [5]. Female and male mouse models of BFLS show an enlargement of lateral ventricles and a propensity for seizures [5].

Epigenetic studies in both BFLS and WHIKERS have shown a characteristic DNA methylation (DNAm) pattern, including a diagnostic episignature biomarker, with a high degree of overlap [7].

Overlapping sensitive and specific (distinctive) signatures for these two syndromes have been generated and can be used in a clinical setting which offers the potential for a more accurate diagnosis and reclassification of VUS. This report demonstrates the clinical utility of this testing.

## 2. Materials and Methods

### 2.1. Genome Sequencing (GS) and Analytic Pipeline

Genome sequencing was performed at the Translational Genomics Research Institute, part of City of Hope, in Phoenix, Arizona. Blood was drawn from the proband, mother, and father, then sent to the TGen Clinical Lab for DNA extraction. Extracted DNA was transferred to TGen’s Collaborative Sequencing Center for library preparation. See the paper by Jepsen et al. for a detailed description of sequencing kits, platforms, and analytical pipeline [8]. Aliquots of DNA were sent from the TGen Clinical Lab to GeneDx for Sanger confirmation of all three variants.

### 2.2. DNA Methylation Episignature Analysis

DNA was extracted from whole blood from the proband. A DNAm profile was obtained using Illumina Infinium MethylationEPIC BeadChip microarray (San Diego, CA, USA) as previously described [9]. Methylation analysis was conducted using the clinically validated EpiSign assay following previously established methods [9,10,11,12]. Methylated and unmethylated signal intensities generated from the EPIC V2 arrays were imported into R 4.2.1 for normalization, background correction, and filtering. Beta values were calculated as a measure of methylation level, ranging from 0 (no methylation) to 1 (complete methylation), and processed through the established support vector machine (SVM) classification algorithm for EpiSign™ disorders. The genome-wide DNA methylation profile of the proband was compared to controls and individuals previously confirmed to have BFLS and WHIKERS.

## 3. Results

### 3.1. Clinical Case History

A 2-year and 9-month-old male presented for evaluation of developmental delay. He was born full term by caesarian section due to breech presentation. The patient’s mother had a history of five miscarriages thought to be secondary to a coagulation defect. He had feeding issues with onset at birth resulting in poor weight gain until six months of age. He developed retching and gagging with solid foods and continues to have difficulty with intake of solids due to sensory issues. He sat at six months, crawled at 17 months, and walked at 28 months. He is nonverbal and displays autism-like behaviours such as hand flapping and lack of eye contact. He had bilateral cryptorchidism, now post-orchiopexy, and bilateral inguinal hernia post repair. His facial dysmorphism includes deep-set eyes, prominent epicanthal folds, telecanthus, large ears, a short nose with a wide nasal tip, hypoplastic alae nasi, a small mouth, and a short neck (Figure 1A). He has pectus excavatum, syndactyly of toes 2–3 bilaterally, fifth digit clinodactyly bilaterally, and tapered fingers. An MRI scan of the brain and total spine was carried out at age 4 4/12 years. Significant findings include a slight asymmetry of the lateral ventricles (left larger than right, at the atrium), focal protrusion of the left anterior temporal lobe through a sphenoid wing bony defect (possible cephalocele), and abnormal T2/FLAIR hyperintensity in the periventricular and subcortical white matter (Figure 1B) The spine MRI was normal except for a slightly thickened filum. He has a younger sister showing neurotypical development. His parents both had learning disabilities as children. At age 38, the mother was 5′ 4″ tall. She had a history of attention deficit disorder as well as reading and comprehension delays as a child. She also has a history of hypothyroidism, soy allergy, with chronic abdominal pain and bloating. She has experienced 5 spontaneous miscarriages. There is no known family history of cancer.

### 3.2. Genomic Testing

Chromosomal microarray showed a duplication on 9p, arr[hg19] 9p24.1 (5108833-5221708) × 3, that included the *JAK2* and *INSL6* genes, classified as a variant of uncertain significance and not thought to be related to the patient’s developmental delay. Heterozygous germline or somatic mutations in *JAK2* are associated with thrombocythemia, while variants in *INSL6* are not known to be associated with a Mendelian disorder.

Family trio whole genome sequencing in a research lab identified the following three variants:(1)A de novo, likely pathogenic heterozygous variant in *SPAST* (c.1413+3_1413+6del, NM_014946.3);(2)A de novo, variant of uncertain significance in *DDB1* (c.344 C>T, NM_001923.4, p.P115L);(3)A maternally inherited, hemizygous variant of uncertain significance in *PHF6* (c.763_765del, NM_032458.2, p.T255del). (See Table 1).

The *SPAST* variant (c.1413+3_1413+6del) is listed in ClinVar (ID 536445) and classified as likely pathogenic or pathogenic (LP/P) [13]. The variant is well-described, having been published in over eight journals and seen in multiple cases of autosomal dominant hereditary spastic paraplegia (HSP) [14,15,16,17,18,19,20]. The variant is not present in population databases (gnomAD v4.1.0) [3]. The SpliceAI score was 0.99, indicating a high probability that the variant affects splicing.

**Table 1 genes-16-00933-t001:** Variants identified in proband’s genome sequencing; * gnomAD v4.1.0.

Gene Name	Variant	Inheritance	Zygosity	ClinVar ID	ACMG Classification	Population Frequency *	Classification	Clinical Significance
*PHF6*	c.763_765del, NM_032458.2, p.T255del	X-linked Maternal	Hemizygous	438300	VUS	Absent	CADD 20.6	Identified in one other individual with ID and dysmorphic facial features [21]. Clinical overlap in the proband includes intellectual disability, undescended testes, and large ears.
*DDB1*	c.344 C>T, NM_001923.4, p.P115L	Autosomal Dominant de novo	Heterozygous	-	VUS	Absent	CADD 24.6, REVEL 0.35, BayesDel score of 0.13, and an AlphaMissense score of 0.581	Clinical overlap in the proband includes developmental delay, large forehead, large ears, differences in his eyebrows (sparse eyebrows spaced farther apart), full cheeks, deep set eyes, epicanthal folds, telecanthus, hypoplastic alae nasi, and 2–3 toe syndactyly.
*SPAST*	c.1413+3_1413+6del	Autosomal Dominant de novo	Heterozygous	536445	Pathogenic	Absent	SpliceAI 0.99	Seen in multiple cases of autosomal dominant hereditary spastic paraplegia (HSP) [14,15,16,17,18,19,20]. Proband does not display signs of spastic paraplegia.

The *DDB1* variant (c.344 C>T, NM_001923.4, p.P115L) is not reported in the literature and has a CADD score of 24.6, REVEL score of 0.35, a BayesDel score of 0.13, and an AlphaMissense score of 0.581. This variant is absent in the population databases (gnomAD v4.1.0); however, a heterozygous individual with a similar amino acid change (p.P115R) is reported within the age range of 50–55 years (unknown phenotype) [3].

The *PHF6* variant (c.763_765del, NM_032458.2, p.T255del) is absent in the population databases (gnomAD v4.1.0) and has a CADD score of 20.6. The variant is listed in ClinVar (ID 438300) and is classified as a variant of uncertain significance [3,13]. The variant was identified in a patient who was part of the CAUSES study [21]. We contacted the individual’s physician, and he described the child’s phenotype at age 6 years as the following: mild-moderate intellectual disability, hyperphagia and obesity, hypotonia, wide-based gait, cryptorchidism, strabismus, dysmorphic facial features (mild synophrys, short forehead, triangular face), fifth finger clinodactyly, and a short penis.

Targeted EpiSign testing (v5) for Börjeson–Forssman–Lehmann, Chung–Jansen, and White–Kernohan syndromes was performed at the London Health Sciences Centre. The patient’s profile was more consistent with BFLS than WHIKERS, but WHIKERS could not be confidently ruled out (Figure 2A–F).

## 4. Discussion

This case is an example of the successful integration of multiple testing modalities (genome sequencing and epigenetic testing) in the diagnostic evaluation of a child with developmental delay. Genome sequencing identified two variants of uncertain significance in genes linked to neurodevelopmental syndromes with an overlapping phenotype (BFLS and WHIKERS). Epigenetic testing helped clarify the clinical significance of these VUS and the diagnosis, impacting medical management. He was also found to have an incidental finding of an LP/P variant in the *SPAST* gene that causes HSP, a disorder for which he currently shows no signs and symptoms.

As in previously reported cases of WHIKERS, the *DDB1* variant (p.P115L) is a de novo missense variant located within the MMS1 domain of the protein. In silico prediction algorithms have conflicting interpretations of pathogenicity, and the variant is absent in population databases. There is some overlap in phenotype for WHIKERS and our patient: developmental delay, large forehead, large ears, differences in his eyebrows (sparse eyebrows spaced farther apart), full cheeks, deep-set eyes, epicanthal folds, telecanthus, hypoplastic alae nasi, and 2–3 toe syndactyly. Without having additional affected patients with this variant or functional studies in animal or cell models, the pathogenicity of the variant remained uncertain.

The *PHF6* variant (p.T255del), while not present in the public databases, has been reported before in another patient with developmental delay and dysmorphic facial features [21]. Again, our patient has phenotypic overlap with BFLS: intellectual disability, undescended testes, and large ears. While the literature reports that heterozygous females can be affected, our patient’s mother, who is heterozygous for the *PHF6* variant, shows no obvious signs of BFLS. Studying the PHF6 protein levels in cell lines from our patient may lead to inconclusive results, as other deletions (p.D333del, p.E337del) have shown normal or even elevated protein levels when compared to controls [5].

Previous epigenetic studies in BFLS and WHIKERS demonstrate a significant overlap in DNAm patterns [7]. A combined methylation study of BFLS and WHIKERS in our patient (Figure 2A,B) provides strong support for the diagnosis consistent with at least one of these conditions. When episignature biomarkers, unique methylation patterns specific for each disorder are used, the results suggest a higher likelihood of BFLS over WHIKERS in the patient, as indicated by comparison of his methylation profiles to those of affected individuals and controls (Figure 2C–F). Given the unique and robust DNA methylation episignature associated with Borjeson–Forssman–Lehmann syndrome and White–Kernohan syndrome, the presence of these disorder-specific epigenetic profiles provides strong functional evidence for pathogenicity. In the context of the *PHF6* variant classification, detection of this matching, well-validated episignature fulfills the ACMG PS3 criterion, thereby supporting reclassification of the variant from VUS to likely pathogenic. An additional secondary diagnosis of WHIKERS cannot be entirely ruled out. Based on the support for a diagnosis of BFLS, additional endocrinology evaluation and imaging studies are recommended. There is also potential for an increased risk of hematological malignancy requiring ongoing surveillance. These results also have further implications for the subject’s family. The proband’s sister has a 50% risk of carrying the variant and may want to consider preconception screening for the variant. Cascade testing may also be appropriate for additional maternal family members with suspected developmental delay or those with reproductive risk.

The de novo *SPAST* variant (c.1413+3_1413+6del) has been reported in multiple patients with HSP with an age of onset starting in the 20s to 50s and presenting as both pure and complicated (additional neurological symptoms) HSP [14,15,16,18]. c.131C>T and c.134C>A are thought to be age of onset modifiers in *SPAST*, leading to very early onset of symptoms. He was not found to have these modifier variants [22]. *SPAST* variants are inherited in an autosomal dominant manner with near complete penetrance [20]. After receiving this result, the patient had a thorough evaluation by his neurologist who did not currently observe signs of HSP. Although our patient shows no signs of HSP, he may develop symptoms later in life, though the severity of symptoms cannot be predicted. While not related to his developmental delay, being able to anticipate symptoms of HSP and prepare for treatments and therapies will be important for the family.

## 5. Conclusions

Here we present a complicated case of a male child diagnosed with developmental delay who had VUSs in two genes associated with disorders with a similar phenotype and an LP/P variant in a gene for which he is presymptomatic. Using a combination of multiple testing modalities (whole genome sequencing and targeted epigenetic testing), we were able to provide valuable insight into the likely cause of his symptoms. Methylation testing showed a strong correlation with other individuals with BFLS, suggesting that this patient’s phenotype is a result of his *PHF6* variant. The testing could not definitively rule out a diagnosis of WHIKERS, so it is still possible (given his physical phenotype) that both the PHF6 variant and the DDB1 variant contribute to his phenotype. In addition, the incidental identification of a previously reported variant in the *SPAST* gene added an unexpected layer of complexity to his case, requiring additional neurological assessments and continued evaluation. This case exemplifies that while functional studies may not be possible in the clinical setting, episignature analysis can provide additional evidence to help solidify a diagnosis and provide families with clarity. In addition, the confirmation of a diagnosis of BFLS guides family planning and may guide his sister and other maternal relatives with targeted preconception screening.

## Figures and Tables

**Figure 1 genes-16-00933-f001:**
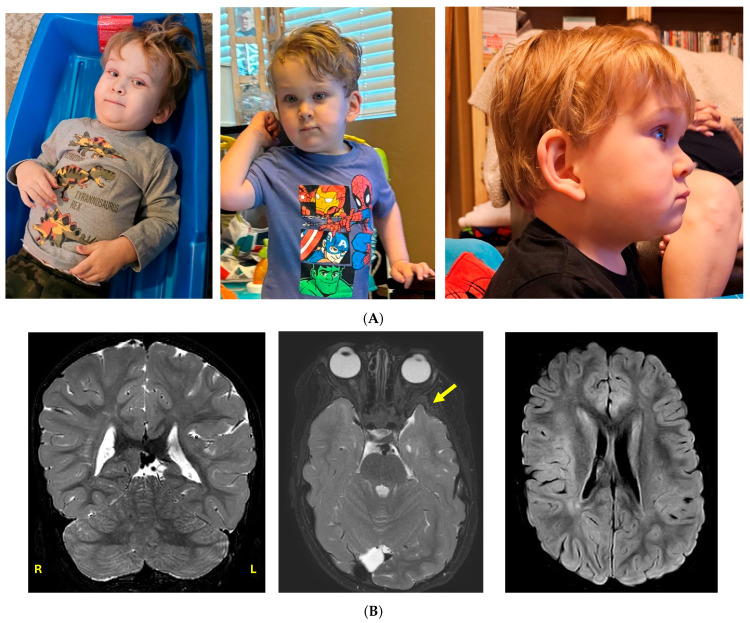
(**A**). Images of proband at 2 years 9 months. Facial features include deep-set eyes, epicanthal folds, telecanthus, large ears, fleshy ear lobes, a short nose with a wide nasal tip, hypoplastic alae nasi, and a small mouth. (**B**) Selected images from an MRI scan of the brain performed at 4 4/12 years of age. Coronal T2 image shows a mild asymmetry of the lateral ventricles (atria), with left being slightly larger than right; axial T2 shows a focal protrusion of the left anterior temporal lobe (arrow) through a presumed bony defect in the sphenoid wing; axial FLAIR image shows signal hyperintensities in the subcortical white matter.

**Figure 2 genes-16-00933-f002:**
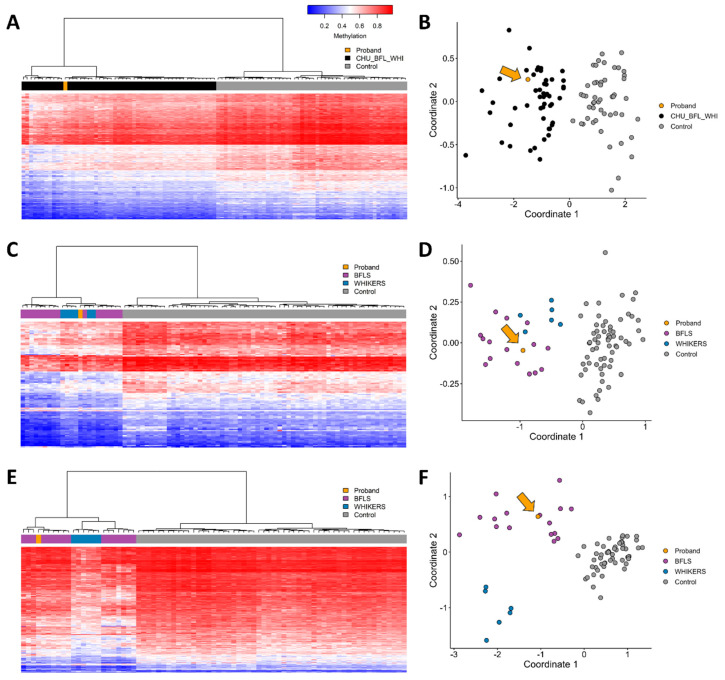
EpiSign analysis of peripheral blood from a proband with variants in *DDB1* and *PHF6*. Hierarchical clustering and multidimensional scaling plots are shown for each episignature analysis. (**A**,**B**) Evaluation using a combined episignature with probes selected to be sensitive and specific for BFLS, WHIKERS, and Chung–Jansen syndrome cases. The proband (orange) clusters with affected cases (black) and is distinct from controls (grey). (**C**,**D**) Evaluation of the BFLS episignature with probes selected using BFLS cases only (purple). WHIKERS cases (blue) show partial overlap, while the proband (orange) clusters preferentially with BFLS cases. (**E**,**F**) Evaluation of the WHIKERS episignature with probes selected using WHIKERS cases only (blue). The proband (orange) shows consistent clustering with BFLS cases (purple). In (**B**,**D**,**F**) orange arrow points to the proband (orange circle).

## Data Availability

The data presented in this study are available on request from the corresponding author due to privacy reasons.

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
