# Peer review of "Integration of Genome and Epigenetic Testing in the Diagnostic Evaluation of Developmental Delay: Differentiating Börjeson–Forssman–Lehmann (BFLS) and White–Kernohan (WHIKERS) Syndromes"

_genes, 2025, doi:10.3390/genes16080933_

Round 1
Reviewer 1 Report
Comments and Suggestions for Authors
Dear authors, congratulations on your manuscript! Your case report is well described, and the application of cutting-edge omics technologies is very impressive.
I have only minor recommendations:
1. A table summarizing the three reported variants would be a nice addition to the text. You can include information about gene name, variant, inheritance, ClinVar ID, zygosity, classification, population frequency, in silico predictions, clinical significance, etc.
2. Can you use epigenetic testing to support reclassification of the VUS, As this type of analysis may qualify as functional evidence under ACMG guidelines?
3. Did you perform deep phenotyping of the mother? Any cognitive assessments? Does she have history of cancer, given the association of PHF6 variants with hematological malignancies?
Author Response
Reviewer 1:
- A table summarizing the three reported variants would be a nice addition to the text. You can include information about gene name, variant, inheritance, ClinVar ID, zygosity, classification, population frequency, in silico predictions, clinical significance, etc.
Table 1 has been provided, and has been cited in the text (line number LL 162).
Table 1 has been included in the revised manuscript and attached separately as an Excel file and as a pdf file.
2. Can you use epigenetic testing to support reclassification of the VUS, As this type of analysis may qualify as functional evidence under ACMG guidelines?
Sentences have been added regarding the functional significance of episignature testing and variant classification under ACMG guidelines. Following text has been added between lines LL 222 and 227.
Given the unique and robust DNA methylation episignature associated with Borjeson-Forssman-Lehmann syndrome and White-Kernohan syndrome, the presence of these disorder-specific epigenetic profiles provides strong functional evidence for pathogenicity. In the context of the PHF6 variant classification, detection of this matching, well-validated episignature fulfills the ACMG PS3 criterion, thereby supporting reclassification of the variant from VUS to likely pathogenic.
3. Did you perform deep phenotyping of the mother? Any cognitive assessments? Does she have history of cancer, given the association of PHF6 variants with hematological malignancies?
We have not been able to reach the mother for a comprehensive interview and examination for deep phenotyping. The following information has been added about the mother and the known family history. Inserted in Lines LL 141-145, highlighted in the revision.
At age 38, the mother was 5’ 4’’ tall. She had a history of attention deficit disorder as well as reading and comprehension delays as a child. She also has a history of hypothyroidism, soy allergy, with chronic abdominal pain and bloating. She has experienced 5 spontaneous miscarriages.
There is no family known family history of cancer.
Reviewer 2 Report
Comments and Suggestions for Authors
To the Authors
In their Ms ID genes-3781374- “Integration of Genome and Epigenetic Testing in the Diagnostic Evaluation of Developmental Delay: Differentiating Börjeson-Forssman-Lehmann (BFLS) and White-Kernohan (WHIKERS) syndromes”- the Authors aimed to demonstrate the clinical utility of epigenetic signatures in prioritizing variants of uncertain significance in genes associated with developmental delay (DD) and intellectual disability (ID). A large (<1500) number of genes and their variants of uncertain significance (VUSs) are known to be associated with developmental delay (DD) and intellectual disability (ID). In the present study, epigenetic signatures successfully differentiated Börjeson-Forssman-Lehmann syndrome (BFLS) from White-Kernohan syndrome (WHIKERS) in a 2 year 9-month-old male proband who was non-verbal, had developmental milestones delay, ASD-like features, and craniofacial dysmorphisms, skeletal abnormalities, and bilateral cryptorchidism. There was a family history of a younger sister with neurotypical development, while both parents had learning disability in their pediatric history. The proband had VUSs in two genes associated with disorders with a similar phenotype (BFLS vs WHIKERS). The epigenetic signatures successfully differentiated BFLS from WHIKERS in this complex clinical case. This information poses the proband at high risk for hereditary spastic paraplegia (HSP) notwhithstanding his current asymptomatic/presymptomatic status. Additional specific endocrinological evaluation and follow-up/surveillance (i.e., increased risk for hematological malignancy) are recommended. The hereditary pattern can lead to preconception screening for the variant in the sister and a cascade testing for additional maternal family members with suspected DD or reproductive risk. The present study indicates that epigenetic testing for disorders with distinct methylation patterns can guide differential diagnostics, as well as genetic counseling / family planning. The case is overall well described and the Ms has certainly merits in terms of novelty/originality and clinical application. I would suggest the Authors address a few minor issues.
Minor Issues
LL123 “caesarian section for breech presentation” --> Please, change into: “caesarean section due to breech presentation”
LL 133-134: “He has a 133 typical younger sister who was meeting her milestones.”--> Please, change into: “He had a younger sister showing a neurotypical development” or a similar statement.
Fig 1. The figure could be complemented with some other imaging (photography/x-rays) of the other skeletal dysplasias reported in the proband.
Author Response
LL123 “caesarian section for breech presentation” --> Please, change into: “caesarean section due to breech presentation”
Changed; highlighted in revised manuscript.
LL 133-134: “He has a 133 typical younger sister who was meeting her milestones.”--> Please, change into: “He had a younger sister showing a neurotypical development” or a similar statement.
Changed; now line number LL 139; highlighted in the revised manuscript.
Fig 1. The figure could be complemented with some other imaging (photography/x-rays) of the other skeletal dysplasias reported in the proband.
The legend to Figure 1 has been expanded to emphasize facial features (Figure 1-A) and includes MRI images (Figure 1-B).
No skeletal series X-rays are available. A recent MRI of the brain and spine were done, and selected images are included in Figure 1-B. The findings on the MRI are also delineated in the revised manuscript text at lines LL 134 to 139.